# SAMBA: Structure-Learning of Aquaculture Microbiomes Using a Bayesian Approach

**DOI:** 10.3390/genes14081650

**Published:** 2023-08-19

**Authors:** Beatriz Soriano, Ahmed Ibrahem Hafez, Fernando Naya-Català, Federico Moroni, Roxana Andreea Moldovan, Socorro Toxqui-Rodríguez, María Carla Piazzon, Vicente Arnau, Carlos Llorens, Jaume Pérez-Sánchez

**Affiliations:** 1Institute of Aquaculture Torre de la Sal (IATS), Consejo Superior de Investigaciones Científicas (CSIC), 12595 Ribera de Cabanes, Spain; fernando.naya@iats.csic.es (F.N.-C.); federico.moroni@csic.es (F.M.); socorro.toxqui@csic.es (S.T.-R.); carla.piazzon@csic.es (M.C.P.); 2Biotechvana, Parc Científic Universitat de València, 46980 Paterna, Spain; ahmed.hafez@biotechvana.com (A.I.H.); roxana.andreea.moldovan@gmail.com (R.A.M.); carlos.llorens@biotechvana.com (C.L.); 3Institute for Integrative Systems Biology (I2SysBio), Universitat de Valencia and CSIC (UVEG-CSIC), 46980 Paterna, Spain; vicente.arnau@uv.es; 4Health Research Institute INCLIVA, 46010 Valencia, Spain; 5Bioinformatics and Biostatistics Unit, Principe Felipe Research Center (CIPF), 46012 Valencia, Spain; 6Foundation for the Promotion of Sanitary and Biomedical Research of the Valencian Community (FISABIO), 46020 Valencia, Spain

**Keywords:** Bayesian networks, metagenomics, machine learning, farmed fish, gilthead sea bream

## Abstract

Gut microbiomes of fish species consist of thousands of bacterial taxa that interact among each other, their environment, and the host. These complex networks of interactions are regulated by a diverse range of factors, yet little is known about the hierarchy of these interactions. Here, we introduce SAMBA (Structure-Learning of Aquaculture Microbiomes using a Bayesian Approach), a computational tool that uses a unified Bayesian network approach to model the network structure of fish gut microbiomes and their interactions with biotic and abiotic variables associated with typical aquaculture systems. SAMBA accepts input data on microbial abundance from 16S rRNA amplicons as well as continuous and categorical information from distinct farming conditions. From this, SAMBA can create and train a network model scenario that can be used to (i) infer information of how specific farming conditions influence the diversity of the gut microbiome or pan-microbiome, and (ii) predict how the diversity and functional profile of that microbiome would change under other variable conditions. SAMBA also allows the user to visualize, manage, edit, and export the acyclic graph of the modelled network. Our study presents examples and test results of Bayesian network scenarios created by SAMBA using data from a microbial synthetic community, and the pan-microbiome of gilthead sea bream (*Sparus aurata*) in different feeding trials. It is worth noting that the usage of SAMBA is not limited to aquaculture systems as it can be used for modelling microbiome–host network relationships of any vertebrate organism, including humans, in any system and/or ecosystem.

## 1. Introduction

Gut microbiomes in fish and other vertebrates are subjected to complex and dynamic fluctuations that are driven by several factors associated with the host (e.g., genotype, physiological status, pathobiology) and its environment, lifestyle and diet [1]. In turn, each of these factors can contribute to improve the sustainability of industrial aquaculture [2]. Hence, the complex relationships between the physical and biological components of aquaculture systems in the context of climate change and human population growth are one of the key future challenges in animal food production [3,4]. Current research on gilthead sea bream (*S. aurata*), a highly cultured species in the Mediterranean, indicated that the gut microbiota is a reliable criterion to evaluate the success of selective breeding with changes in diet composition [5,6,7]. However, our understanding about these kinds of dynamics is at an infancy state, due to their inherent complexity, the multiple biotic and abiotic factors involved, and the enormous variability of mucosal microbial populations among distinct individuals of the same population [8,9]. For this reason, there is a growing interest to develop tools that can model how fish microbiomes and their hosts interact under variable farming conditions. Along these lines, Bayesian networks (BN) and structure learning [10,11,12] may be especially useful due to their capacity to infer directional relationships in microbial communities [13,14]. Certainly, BNs are probabilistic graphical models based on the Bayes Theorem that represent and evaluate the conditional dependencies among a set of variables via directed acyclic graphs (DAG). In such models, variables and their interrelations of dependency are represented as nodes and edges, respectively [15]. Structure learning refers to the process to learn the structure of the DAG from the available data, creating a model where an edge between two nodes indicates direct stochastic dependency, while no connection (edge) between two nodes identifies that the corresponding variables are independent or conditionally independent [13].

BNs have been used to promote the sustainable development of aquaculture systems [16]. However, they are yet to be applied to reveal dynamic interactions between different biotic and abiotic factors. Moreover, BN tools are typically tailor-made solutions created using command line interface (CLI) software packages (e.g., bnlearn), but they are complex to manage and often restricted to expert bioinformaticians and computational biologists [17]. Indeed, most user-friendly BN tools with Graphical User Interfaces (GUI), such as ShinyBN [18] or BayesiaLab [19], only work with discrete variables and small datasets. In addition, while the recently released BayeSuites tool [11] manages continuous variables and large datasets, it currently cannot make inferences and establish conditional probability distributions based on discrete variables. To address these issues, we created SAMBA (Structure-Learning of Aquaculture Microbiomes using a Bayesian Approach), a new BN tool to investigate microbiome systems or microbiome–host network dynamics in aquaculture systems by modelling how fish gut microbiomes and/or pan-microbiomes interact with the various biotic and abiotic factors. Here, we provided examples of the functionality of the tool’s web-interface, and evaluated SAMBA performance when building and estimating the conditional dependencies among the variables in the DAG model. To this end, we will use two training datasets of different natures and complexities: i) an artificial microbial community with few taxa and defined composition; and ii) real fish microbial communities of *S. aurata* resulting from a given time and aquaculture infrastructure, with diet as the main changing experimental variable. This first approach with SAMBA is fundamental not only from a testing point of view, but also to “feed” the tool with complex datasets. These experimental data will compose a reservoir of information that will make SAMBA a reliable predictive tool for investigating changing and sometimes poorly predictable scenarios.

## 2. Materials and Methods

### 2.1. SAMBA Modules

SAMBA is currently available in a GitHub public repository. The URL for downloading the input data is reported in the Data Availability Statement. The tool can be installed on personal computers, and it is based on a backend engine core that consists of a set of workflows and pipelines implemented in R and Python using third-party software dependencies. The frontend component of SAMBA consists of a web-based Graphical User Interface (GUI) implemented using shiny [20] to provide a friendly and intuitive interface to manage the engine core. Functions and tasks of SAMBA are structured in five modules: “Build”, “Inference”, “Prediction”, “Viewer”, and “Downloads”. A User Guide including technical details about the algorithmic basis of SAMBA is provided in Appendix A.

“Build”: This module creates and trains BN models from the provided input data using a pipeline based on the *bnlearn* R package [17]. This module works with continuous and discrete variables. However, a discretization optional step is implemented using one of the following methods: interval, quantile or Hartemink [21]. For those continuous variables that are not discretized, a Shapiro [22] test is performed to know if they follow a normal distribution. For further information, please refer to Appendix A. To create the BN models, SAMBA allows the user to fit distribution parameters. The current implementation includes a normal distribution in a logarithmic scale (Log-Normal) and a generalized linear model, the Zero-inflated Negative Binomial (ZINB) distribution, that better fits highly dispersed data with an excess of zeros in the taxa abundance counts [23]. The hc() function of bnlearn learns the structure of a BN using a hill-climbing (HC) greedy search (score-based algorithms). According to Scutari et al. (2019) [24], these algorithms are usually faster and more accurate for both small and large sample sizes. The HC search [25] explores the space of the DAG by single-arc addition, removal, or reversals. It also assigns a rate to the BN model using catching, decomposability, and three equivalence score functions to reduce the number of duplicated tests [17]. The three score functions are the Akaike Information Criterion (AIC), Bayesian Information Criterion (BIC), and Multinomial log-likelihood (loglik). The score method is adjusted for the hc() method so that a higher score is generally preferred.

The training of the model constructed by the “Build” module is performed by using the bn.fit() function of bnlearn or the aforesaid bn.fit() combined with the zeroinfl() function from the pscl package [26]. The bn.fit() function fits, assigns, or replaces the parameters of a BN conditional on its structure, while zeroinfl() fits zero-inflated regression models for count data via maximum likelihood. Once the parameters have been fitted, the strength of each connection is calculated using BIC and Mutual Information (MI) criterion [27] and the arc.strength() function of bnlearn to remove all links with strength values greater than the user-defined threshold. The future() function of the future package [28] allows the user to continue using other functions in the app while a model is being computed.

“Inference”: This interface uses functions of bnlearn and dagitty [29] to infer how the diversity of the pan-microbiome indexed in the BN is influenced by the experimental variables (season, diet composition, temperature, genetics, etc.). The “Inference” module provides two different report options: conditional probability tables (CPTs) and DAG. The CPT option uses a cosponsoring quantile-quantile plot of the fitted node to show the type of relationships among different taxa under different experimental variables. The DAG option creates a DAG from the bnlearn output using the dagitty() method of dagitty and allows the markov blanket of a given node to be extracted from the DAG using a markovBlanket() method.

“Prediction”: This interface allows the user to manage two predictive pipelines. The first, “Predict abundances” is a workflow based on bnlearn and pscl that predicts how taxa abundance counts will likely change based on changes in one or more farming variables selected as conditional evidence. In the Log-Normal Distribution mode, normalized frequencies in log scale are obtained via the cpdist() function of bnlearn. In the ZINB distribution mode, a custom sampling method samples from the fitted ZINB models of each taxon in the BN. The second pipeline of this module, “Predict Metagenomes”, infers the metagenome of a given pan-microbiome under specific experimental variables. It uses PICRUSt2 [30], which includes two different database annotation protocols: MetaCyc [31] and KEGG [32]. For more details about the PICRUSt2 workflow and its dependencies [33,34,35,36,37] please refer to Appendix A.

“Viewer”: This module provides the user with tools to visualize, edit, customize, navigate, and export the DAG in various graphical formats. The “Viewer” is implemented using commands from several different packages. The subgraph() function of bnlearn plots the graph. VisIgraph() and the renderVisNetwork() functions of visNetwork [38] provide an interactive display of the DAG. The strength.viewer() function of bnviewer [39] shows the strength of the probabilistic relationships of the BN nodes. It also uses model averaging to build a graph containing only significant links. The decompose() function of the igraph package [40] visualizes specific node groups and, thus, allows users to work with a specific subgraph. CPTs with conditional probability information about the inter-relations of each node can be displayed in the viewer using the datatable() and dataTableProxy() functions of the *DT* package [41]. These functions allow the user to browse and filter information from the DAG. The “Viewer” also integrates a sidebar with tools for highlighting, selecting, and/or editing specific nodes and features using functions from the VisNetwork package, such as visOptions(), visRedraw(), visSetSelection(), visUpdateNodes(), and visUpdateEdges(). It also contains JavaScript code introduced through the runjs() function of the *shiny* package [42] and the JS() function of the htmlwidgets package [43]. Graphs can be downloaded as HTML, PNG, JPEG, or PDF files. A screenshot can be taken of the current network and exported using the shinyscreenshot package [44].

“Downloads”: This is a repository for the user to download.zip files containing results and output files from the “Build” module (the output includes normalized counts, link strengths, and a RData file containing the BN model) or the “Prediction” module (provided by metagenomic prediction).

### 2.2. Artificial Testing Dataset (Sequencing and Dataset Definition)

Semi-synthetic bacterial community ZymoBIOMICS™ Microbial Community Standard II (Zymo Research Corp., Irvine, CA, USA) was used as the dataset for constructing network models with SAMBA. This “mock community” is composed of eight bacteria (*Listeria monocytogenes*, *Pseudomonas aeruginosa*, *Bacillus subtilis*, *Salmonella enterica*, *Escherichia coli*, *Lactobacillus fermentum*, *Enterococcus faecalis*, and *Staphylococcus aureus*) with known differential abundances, distributed on a log scale, ranging from 0.00001% (*S. aureus*) to 95.9% (*L. monocytogenes*). PCR conditions and sequencing procedures were performed as described by Toxqui-Rodriguez et al. (2023) [45]. Briefly, eight replicates of the mock community were sequenced using the Oxford Nanopore Technologies MinION platform. The complete 16S rRNA gene (V1–V9) was sequenced using an R9.4/FLO-MIN106 flow cell with the 16S Barcoding Kit 1–24 (SQK–16S024) protocol version 16S_9086_v1_revR_14Aug2019. Reads were then demultiplexed and basecalled using MinKNOW v21.11.17 and Guppy v5.1.12. The resulting reads were preprocessed using Porechop v0.2.4 [46] for adapter removal, NanoFilt [47] for length-filtering between 1200 and 1800 bp, and yacrd [48] for chimera removal. Taxonomy assignment and abundance quantification was performed using Minimap2 [49] aligning sequences against the SILVA database [50]. The two PCR conditions were optimized starting from the recommendations of the kit’s manufacturer: PCR1 (temperature of annealing 55 °C, 25 PCR cycles) and PCR2 (temperature of annealing 52 °C, 30 PCR cycles) [45]. The raw abundance counts (with the exception of *S. aureus*, which was not detected after taxonomic assignment) and the PCR conditions to sequence the mock community were used as input to SAMBA. The mock community was utilized to test the accuracy of the SAMBA modules for BN model construction and prediction based on data with no biological variability. To this end, the BN model was created by the HC algorithm (no limit on the number of iterations, terminated by algorithm convergence) and utilizing all 8 replicates of the mock dataset. After building, the RData of the BN model were used as an input to the “Prediction” pipeline to predict the most likely abundances of each taxon using the log-normal distribution and default parameters. The most likely abundance values were extracted from the prediction module after setting the corresponding values for the PCR experimental variable. Full details about this dataset are provided in Appendix A and the URL for downloading the input data are reported in the Data Availability Statement.

### 2.3. Empirically Testing S. aurata Dataset (Sequencing, Experimental Design, Rearing Conditions, and Dataset Definition)

Intestinal pan-microbiome data were taken from the results of three published experimental trials that used *S. aurata* as a case study host model [51,52,53]. The *S. aurata* dataset included 844 taxa classified at the genus level that were obtained by sequencing the V3-V4 hypervariable regions of 16S rRNA, using the Illumina MiSeq system (2 × 300 paired-end run) (Illumina Inc., San Diego, CA, USA) at the Genomics Unit from the Madrid Science Park Foundation (FPCM, Campus de Cantoblanco, Spain). The autochthonous microbiota populations were sequenced from both the anterior and/or posterior intestine sections of 72 randomly selected specimens of *S. aurata*. Briefly, the trials were conducted in parallel (spring–summer 2020) at the Institute of Aquaculture Torre de la Sal under natural light and temperature conditions (40°5′ N; 0°10′ E), using fish with the same genetic background (sibling animals from the same hatchery batch; Avramar, Burriana, Spain). The resulting microbial populations were investigated in relation to different feeding scenarios (LSAQUA, EGGHYDRO, GAIN_PRE), which are summarized in Table 1. In the LSAQUA trial, fish meal (FM) was either partially (50%) or completely (100%) substituted with a protein replacer of processed animal proteins (PAPs) and bacterial single-cell proteins (SCPs). In the EGGHYDRO trial, combinations of FM and fish oil (FO) were used with or without a bioactive egg white hydrolysate. In the GAIN_PRE trial, FM was completely replaced with alternative protein sources (aquaculture by-product meal, insect meal, microbial biomass, and plant proteins) supplemented with a commercially available health-promoting feed additive. The performance of SAMBA was tested using the *S. aurata* dataset to build and train a BN model under the following parameters: BN score (BIC), taxa distribution (ZINB), link strength thresholds (MI < 0.05; BIC < 0), and a prevalence filter of 50 (which dismiss those taxa that are present in less than 50% of the samples with at least a minimum abundance of 1 read count). The experimental variables (Table 1) were considered independent so that predictions could be made for scenarios that were not directly tested in the methodology. The BN model was created by the HC algorithm (no limit on the number of iterations, terminated by algorithm convergence), utilizing 72 samples in the dataset. The predictive performance of SAMBA was evaluated in the EGGHYDRO trial. In addition, to further assess the predictive function of SAMBA, we tested how the modification of a variable condition, specifically FM, would affect the microbiome abundances of EGGHYDRO trial Scenario 3. This virtual Scenario, called as Scenario 4, was designed with the following experimental variables: FM > 20; FO ≤ 4; “AI” Tissue; “EWH” additive. The *S. aurata* dataset can be downloaded as reported in the Data Availability Statement.

## 3. Results and Discussion

Modelling the complex relationships of physical and biological components are particularly relevant for fish farming because the dynamics and hierarchies of the fish microbiomes and pan-microbiomes can reveal insights into the effects of genetics, environment, or traceability factors. In this article, we introduce SAMBA, the software implementation of a BN approach to learn, build, and train BN models from input datasets with quantitative and qualitative variables (including taxa abundance raw counts). As shown in Figure 1, SAMBA has a user-friendly GUI interface that provides access to five modules (“Build”, “Inference”, “Prediction”, “Viewer” and “Downloads”), which overcomes the usual computational complexity that exists in the modelling of BNs (see Appendix A for technical details). The application can be used to investigate the causal relationships between microbiomes and their hosts by deciphering how the taxa population are related each to other and influenced by the experimental variables. SAMBA can also be used to navigate the built BN model and to inspect the distribution of conditional dependences among the distinct variables, identifying those that provide statistically significant information about how a change in feed formulation, or any other environmental condition, may derive in a modulatory effect in the microbial profile. Additionally, SAMBA conditional BN dependencies provide a system biology perspective that, in comparison to conventional analyses based on the relative abundance of the taxonomic groups, is highly informative because the user may find combined actions between taxa without making independence assumptions, as normally performed in a usual 16S count analysis.

To test the potential and predictive power of SAMBA, we fed the tool with two training datasets (the mock community and the *S. aurata* dataset) to create two BN models. First, given its simplicity and semi-synthetic nature (no natural inter-sample dispersion among the abundance counts of the modelled taxa and only PCR conditions as experimental variable), the mock community offers a controlled scenario for assessing and showing the accuracy of SAMBA for making microbiota predictions. In Figure 2A, we show the DAG resulting from the mock community BN model, which depicts how the seven taxa of this community are connected each to other in the network as a result of their abundances and the experimental PCR condition (with the exception of *L. fermentum*, which is not affected by the PCR condition because it was the least abundant taxon). Predictions based on the mock community BN model were also performed and provided in Appendix A. In addition, Figure 2B,C contains two correlation plots which show a remarkable linearity between the observed and the predicted abundances (under the two PCR conditions) of the seven taxa constituting the mock community. These two analyses are both supported by correlation coefficients over 0.99 for both PCR1 and PCR2 conditions, and with *p*-values for a F-test of 6.12 × 10^−10^ and 1.48 × 10^−22^ in the analysis of variance (see Appendix A). We can, thus, conclude that SAMBA predictions accurately approximate the real-world observation from lab-made microbial simplified scenarios, which could be particularly helpful in designing and exploring synthetic biology experiments.

The second BN approach based on the *S. aurata* dataset was performed to assess how SAMBA builds and manages BN models using microbiome data from real-world experimentation (i.e., with high levels of dispersion in the taxa abundance counts). In Figure 3A, we show the DAG of the BN built using the *S. aurata* dataset. The prevalence filter of 50% reduced the number of taxa to 45. This number represents the core microbiome present in at least 36 of the 72 samples. This is a useful feature of SAMBA as it allows the user to focus not only on the whole dataset or the most abundant OTUs in the bacterial populations (the usual approach of 16S metagenomic analyses), but also on different subsets by managing the filtration parameters of the interface. Extracting functional and quantitative information from the DAG with SAMBA is easy and intuitive with the inference module. In particular, an example of how these results can be obtained is reported in Figure 3B, with the Markov blanket graph extracted from the total DAG. The image shows that the node representing *Pseudomonas* is the child of the nodes representing the experimental variables FO, FM and *Phyllobacterium* that indirectly connects *Pseudomonas* with *Clostridium* sensu stricto. The coefficient for conditional probability distribution that is significant for the node *Pseudomonas*, and the results of a Z test for each coefficient are shown in Appendix A. According to this, it is possible to observe in Figure 3B that the FM variable (in its three states) has a significant impact on the abundance of *Pseudomonas*. In contrast, the variable FO only significantly impacts *Pseudomonas* when FO ≤ 4, but not when 4 < FO < 12. Additional to the dependences that occur between the experimental variables and taxa, the Markov blanket also offers information regarding the taxa–taxa interaction. The presence of *Phyllobacterium* in the pan-microbiome is significant in respect to *Pseudomonas*. This relationship means that the abundance of *Phyllobacterium* influences the abundance of *Pseudomonas.* Nevertheless, the impact of *Phyllobacterium* is less than that of FM and FO, which are the main causes for the variability in abundance of *Pseudomonas* in the *S. aurata* BN model. All these findings make SAMBA a useful tool, which allows the user to go one step forward in the comprehension of the inner dynamics of the microbial community. Regarding this, another example of functional information that can be obtained from the DAG is represented by the edges between OTUs and experimental variables. As the case of *Pseudomonas*, SAMBA detected other causal relationships, which connected other important genera like *Streptococcus*, *Sphingomonas*, *Photobacterium*, *Massilia*, *Corynebacterium* and *Staphylococcus* (Figure 3A) with components of the diet, such as FM, FO, and additives. The identification of which bacteria are more susceptible to changing in a diet and the magnitude by which they change their abundance, represents a cornerstone for nutrition in aquaculture. Defining the links that determine the pan-microbiome community structure when different feed conditions (as the case of the present experiments) or different environmental conditions are applied, provides a powerful forecasting tool to be used to face aquaculture challenges, such as the achievement of a more sustainable production sector through new innovative feed formulations.

The *S. aurata* dataset was also used to test the predictive capability of SAMBA. The profile of taxa abundances was predicted for the three feeding scenarios of the EGGHYDRO trial (Scenarios 1, 2 and 3 in Table 1), using the distribution of probabilities provided by the *S. aurata* BN model. Full predictive reports for Scenarios 1, 2 and 3 are available in Appendix A, which shows that the probability density value of the range of the generated samples (the (P(M+SD)) was above 0.60 for 44 of the 45 taxa in Scenario 1; for 39 of the 45 taxa in Scenario 2; and for 42 of the 45 taxa in Scenario 3. This means that under Scenarios 1, 2 and 3 the ranges of predicted values made by SAMBA are significant for 98%, 87%, and 93% of the taxa, respectively. Moreover, with some reasonable exceptions, the mean, median, standard deviation, and quantiles overlap with the SAMBA predictions (Figure 4 and Figure 5). Another approach that we used to test the predictive performance of SAMBA involved the creation of a confusion matrix. This table, reported in Appendix A, were designed calculating different parameters using the same profile of predicted taxa abundances obtained from the three feeding scenarios of the EGGHYDRO trial (Scenarios 1, 2 and 3 in Table 1) discussed before. The assessment was calculated using a linear regression (Pearson Correlation) between the predicted and observed values, and a True/False Positive and Negative classification. In particular, the evaluation was performed considering True Positive when the predicted value fell within the range defined by the average ± the standard deviation of the real observation, and was instead a False Positive when it was out of that range. On the other hand, the values were considered True Negative when the prediction was 0 and it fell within the range of the real observation, while a False Negative when the prediction was 0 and out of that range. The precision of the measurements and, hence, the tool, was then expressed as the rate between TP and the sum of TP and FP. The results display a strong statistical significance (*p* < 0.001) in the linear regression for all three feeding scenarios, with adjusted R^2^ parameter ranges of 0.69–0.79, 0.61–0.86 and 0.62–0.85. The number of FP predictions detected was minimum and the FN absent, with a precision (calculated as the rate between TP and the sum of TP and FP) very similar for all the scenarios considered (between 0.79–0.83). Thus, we conclude that SAMBA accurately predicts taxa abundances in a large matrix of data, even if their abundance distribution show significant dispersion due to the inter-sample biological variability, like real microbiota data [8,9]. Scenario 4 also tested the predictive capability of SAMBA by combining information from two different trials. More specifically, Scenario 4 predicted how the profile of taxa abundances of Scenario 3 would likely change under the experimental condition of Scenario 1 when FM > 20 (Appendix A). In Figure 5, we also plotted mean, median, standard deviation, and quantiles obtained from Scenario 4 to compare with those from Scenario 3. The figure shows how the predicted abundance of some taxa significantly changed in response to FM. Thus, this suggests that SAMBA allowed the exploration of virtual scenarios using the BN model. This feature of SAMBA is useful for improving both the productivity and sustainability of fish farms as it can simulate, a priori, the effects in the microbiome–host inter-relations under different conditions (e.g., new diet, additives, or a modification of the environmental conditions).

When considered together, our results highlight the capacity of SAMBA to identify which FM or FO dietary levels have a significant influence on the microbiota profile of farmed fish. Understanding these associations emphasizes the ability of SAMBA to predict changes in the microbial communities of *S. aurata* as a case study of aquatic farmed animals. In addition to this, the information that comes from taxa interactions can give interesting information on the dynamics of a collaborative and/or competitive nature that rule a complex environment, such as the intestinal biome. The future use of SAMBA will expand the available testable experimental conditions, allow users to customize their analysis, and introduce their personal expertise and knowledge when modelling a microbe population. SAMBA can, therefore, be a valuable tool for making the research in the aquaculture field more dynamic. In any case, the use of SAMBA is not restricted to fish farming and aquaculture, as it can be adapted to build microbial-host BN models from other systems and/or vertebrate organisms, including humans. In fact, the input dataset accepted by SAMBA only consists of two files: one with the abundance counts per bacterial taxa and another with the experimental/environmental variables. In the present case study, the experimental variables were discrete (ergo categorical). However, the tool is able to manage both continuous and categorical experimental variables, such as sex, age, specimen size, genetic background, tissue, season, diet composition, pH, temperature, phenotypes, and more. Furthermore, in this version of SAMBA, we used a single-omics (16S amplicons originated from Oxford Nanopore Technologies MinION platform as well as Illumina MiSeq system) model to assess microbiome–host network interrelations. However, future updates will integrate multi-omics variables from RNAseq and Methylseq, as well as other layers of information. This will extend the applicability of SAMBA to other topics where BNs have been proven to be effective, such as behavioral and welfare assessments, epigenomics, and genomics and transcriptomics [54,55,56].

Regarding data distribution, the tool is currently limited to two models (Log-normal and ZINB). Future versions will include other generalized linear models such as Negative Binomial, Poisson, and γ [23,57]. Additionally, different normalization methods, including for instance scaling factors for data correction per sample depth and gene size, will be considered (RPKM, FPKM, TPM, RMS, etc.) [58]. We also aim to implement tools for determining the minimum training sample size needed to detect a significant effect for a given experiment [59]. Meanwhile, the first SAMBA release highlighted the potential of this tool as a valid approach to investigate how the experimental variables influence microbial biomes. SAMBA, with a user-friendly interface, allows any user to exploit its full potential (all-in-one solution), without the need to know programming languages and/or combine multiple platforms. The tool deconstructs and quantifies the structure of network relationships affecting the microbial dynamics of a given microbiome dataset and allows us to obtain realistic predictions not only from tested, but also from inferred experimental conditions. Therefore, repetitive SAMBA executions with new datasets and the further implementation of multi-omics data will continually improve the output of this platform, making it a valuable easy-to-use advancement in aquaculture practice for physiologists and nutritionists, as well as fish farmers and breeders.

## Figures and Tables

**Figure 1 genes-14-01650-f001:**
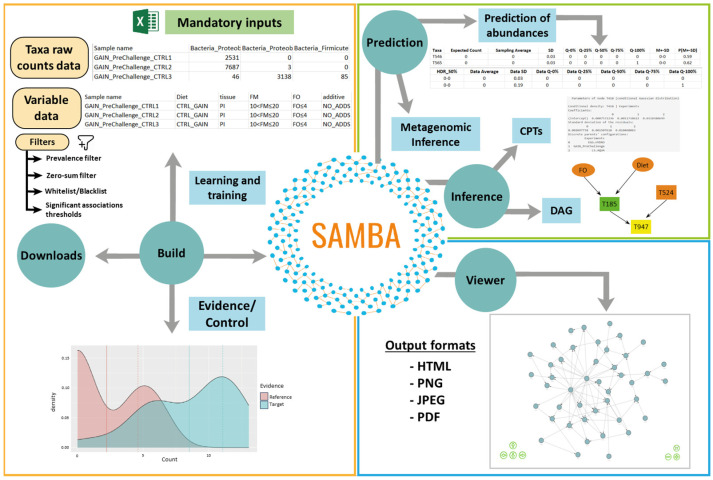
Graphical description of the functions and interface panels in each module of SAMBA. Green circles represent modules and blue squares represent interfaces.

**Figure 2 genes-14-01650-f002:**
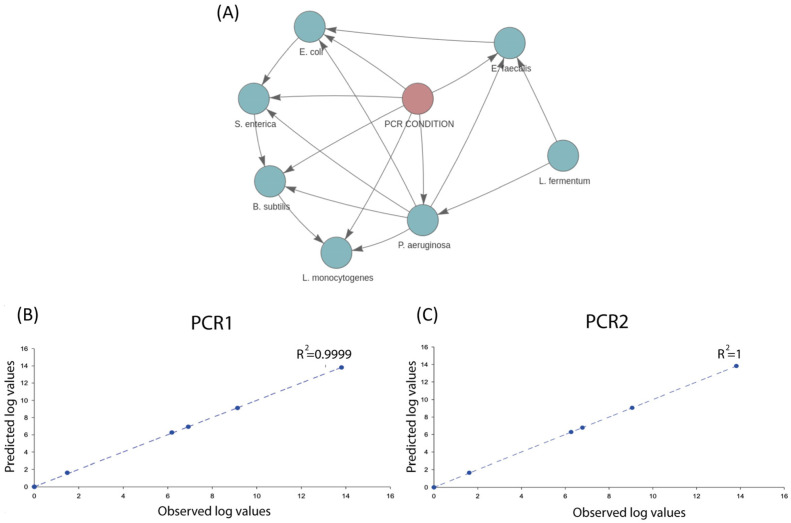
(**A**) Screenshot of the BN model created by SAMBA for the mock community showing how the distinct taxa are related to each other; (**B**) correlation analysis between the average predictions for the seven taxa relative to the normalized average values of abundances, in log scale, under PCR1 condition; (**C**) same correlation analysis but under PCR2 condition.

**Figure 3 genes-14-01650-f003:**
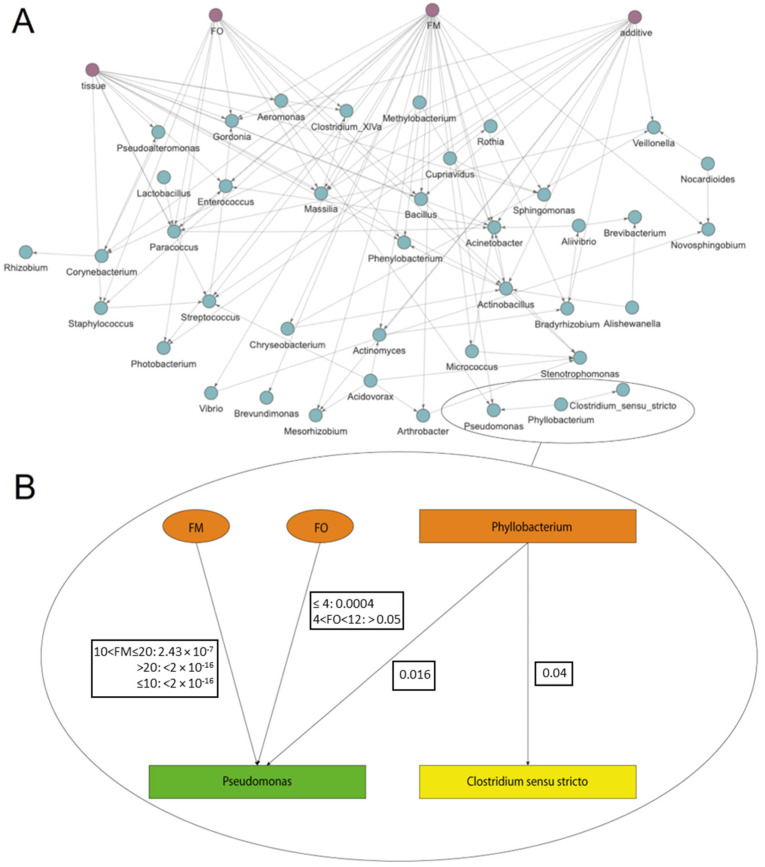
(**A**) *S. aurata* model DAG built with SAMBA using the ZINB distribution showing the significant edges calculated between the experimental variables. Experimental variables are in pink while taxa are blue. (**B**) Markov blanket of *Pseudomonas* node extracted from the total DAG. Taxa are represented with rectangles while experimental variables are represented with ovals. The *Pseudomonas* node is green, nodes with direct relationships with *Pseudomonas* (FM, FO and *Phyllobacterium*) are orange, and nodes with indirect relationships are yellow. The *p* values refer to whether a taxon or a variable has a significant influence (*p* < 0.05) on the *Pseudomonas* node.

**Figure 4 genes-14-01650-f004:**
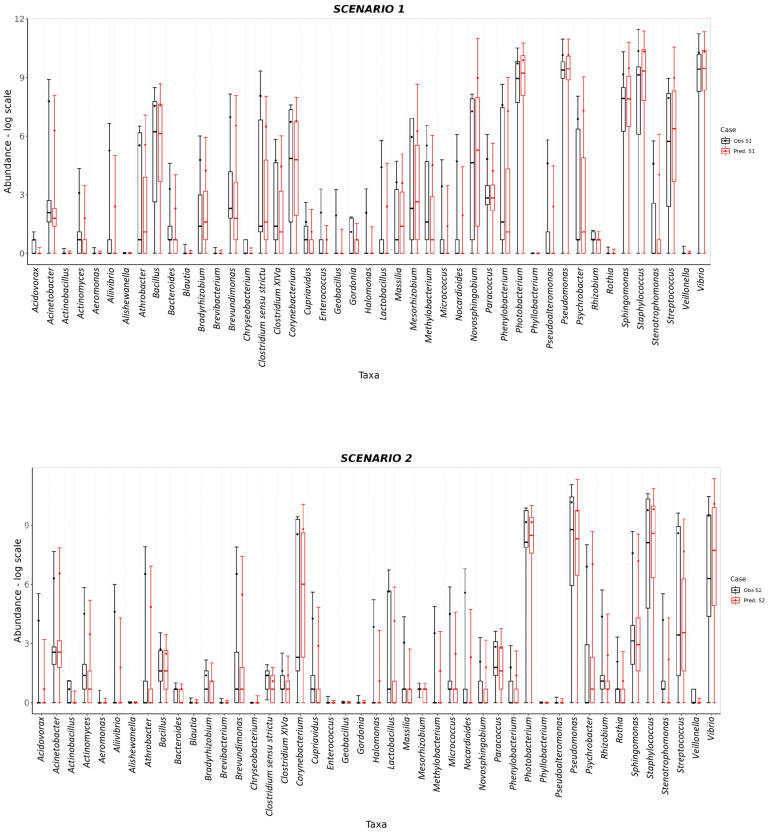
BoxPlot comparisons between predictions (red) and experimental observations (black) in log scale for Scenario 1 and Scenario 2. In both cases, the box plot for each taxon covers a range of values defined by the average abundance for that taxon and its standard deviations. The 25 and 75% quantiles as well as the median abundance values are represented as boxes.

**Figure 5 genes-14-01650-f005:**
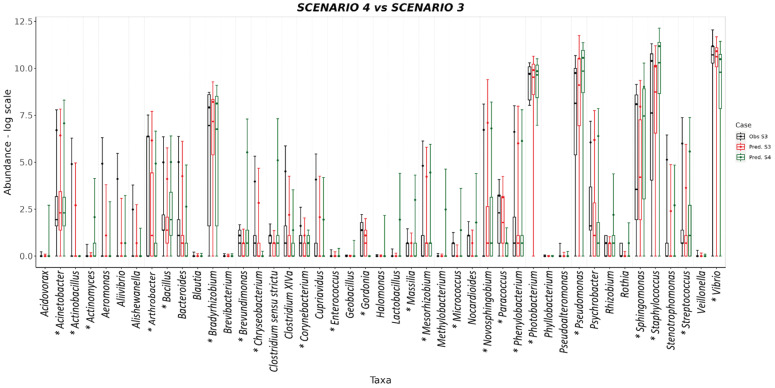
BoxPlot comparisons between predictions and experimental observations in log scale for Scenario 3 and Scenario 4 which is a prediction about how Scenario 3 would likely change when changing the FM conditions. Predictions for Scenario 3 are represented in red and observations black. Predictions for Scenario 4 are represented green. No observed data are provided for Scenario 4 because it is a virtual scenario that is derived from combining experimental condition FM of Scenario 1 with the experimental conditions for FO, TISSUE, and ADDITIVE of Scenario 3. As in Figure 4, each box plot includes information from the average abundance and standard deviations for each taxon plus the quantiles 25 and 75 and the median for the distribution of abundance counts. Taxa that are significantly influenced by variable FM are highlighted with an asterisk (see also Figure 3).

**Table 1 genes-14-01650-t001:** Experimental variables for the three gilthead sea bream farming trials.

Feeding Scenarios	FM	FO	Tissue	Additive/Substitute
LSAQUA				
Scenario 1	10 < FM ≤ 20	4 < FO < 12	AI	NO_ADDS
Scenario 2	≤10	4 < FO < 12	AI	LSAQUA
Scenario 3	≤10	4 < FO < 12	AI	LSAQUA
EGGHYDRO				
Scenario 1	>20	4 < FO < 12	AI	NO_ADDS
Scenario 2	10 < FM ≤ 20	≤4	AI	NO_ADDS
Scenario 3	≤10	≤4	AI	EWH
GAIN_PRE				
Scenario 1	10 < FM ≤ 20	≤4	PI	NO_ADDS
Scenario 2	≤10	4 < FO < 12	PI	SANA

FM = fish meal; FO = fish oil; TISSUE defines the targeted intestinal portion (AI = Anterior; PI = Posterior); ADDITIVE/SUBSTITUTE denotes the existence of an additive or commercial protein replacer (NO_ADDS = without additives; SANA = with SANACORE^®^GM; EWH = with egg white hydrolysate; LSAQUA50 = 50% of FM substitution with LSAqua SusPro^®^; LSAQUA100 = 100% of FM substitution with LSAqua SusPro^®^).

## Data Availability

The source code of SAMBA as well as a dataset for testing the application are available at https://github.com/biotechvana/SAMBA (accessed on 23 June 2023). The two testing datasets (“Mock_community” and “*S. aurata*_dataset”) used to evaluate the performance and accuracy of SAMBA as well as the R.Data file with their respective BN models are available at the Web site of SAMBA with the following URL address https://github.com/biotechvana/SAMBA/tree/main/Testings_datasets (accessed on 23 June 2023). Each dataset includes two files containing the experimental variables (i.e., diet, tissue, additive, etc.), and the raw counts of all taxa per amplicon sample. This URL also includes another dataset named “metagenome_testing_dataset”, which is provided for users interested in training examples for metagenome predictions using the “Prediction” module of SAMBA. Original Fastq files from Mock community, the LSAQUA, EGGHYDRO, and GAIN_PRE trials are available at the SRA archive with the following bioproject accessions PRJNA891255; PRJNA713764; PRJNA705868; PRJNA750446.

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
