# Peer review of "SAMBA: Structure-Learning of Aquaculture Microbiomes Using a Bayesian Approach"

_genes, 2023, doi:10.3390/genes14081650_

Round 1

Reviewer 1 Report

The manuscript titled "SAMBA: Structure-Learning of Aquaculture Microbiomes using a Bayesian Approach" presents a novel computational tool for modeling the network structure of fish gut microbiomes and their interactions with biotic and abiotic variables in aquaculture systems. The study demonstrates the functionality and predictive power of SAMBA using both synthetic and real-world datasets. The manuscript is well-written and provides valuable insights into the application of Bayesian networks in understanding microbiome-host interactions. However, there are several areas that could be improved:

1. The manuscript briefly mentions the potential applications of SAMBA in improving the productivity and sustainability of fish farms. However, it would be beneficial to provide more concrete examples or discuss how SAMBA can be utilized in practical aquaculture scenarios. Highlighting specific challenges or problems in aquaculture that SAMBA can address would make the manuscript more relevant and compelling to the aquaculture community.

2. The article describes the application of SAMBA to two datasets, including a mock community and real fish microbial communities. While the results demonstrate the predictive capabilities of SAMBA, it would be beneficial to include additional validation methods such as cross-validation or comparison with other established modeling approaches. This would enhance the robustness of the findings and provide a more comprehensive assessment of SAMBA's performance.

1. One improvement that is needed in the manuscript is a thorough proofreading and editing pass to address the grammatical errors and unclear sentences. In line 72-73, ' To address these issued, we created SAMBA (Structure-Learning of Aquaculture Microbiomes using a Bayesian Approach). ' The original sentence incorrectly uses "issued" instead of "issues." In line 329-330, ' The figure shows how the predicted abundance of some taxa significantly changed is response to FM.' Suggested revision: ' The figure shows how the predicted abundance of some taxa significantly changed in response to FM.'  In line 318-319, ' This means that under Scenarios 1, 2 and 3 the ranges of predicted values made by SAMBA is significant for 98%, 87%, and 93% of the taxa, respectively. ' The subject "ranges" is plural, so the verb should also be plural. Therefore, "is" should be changed to "are."

Reviewer 2 Report

Reviewer comments

Genes

Title: SAMBA Structure-Learning of Aquaculture Microbiomes using a Bayesian Approach

General comments

The manuscript titled “SAMBA Structure-Learning of Aquaculture Microbiomes using a Bayesian Approach” is an interesting research. Authors attempted to use microbiomes data to establish a predictive model for agriculture or aquaculture. It was a challenging task for the authors to collect a massive amount of data to build the model. It is a good attempt and could contribute to agriculture. SAMBA can predict microbiome information using simple data. Although the results of this study are interesting, they may lack detailed descriptions. More information should be added to this study. In the Materials and Methods section, it is crucial to provide a comprehensive description of the collection sources for all datasets, including the location, method of collection, quantity of samples, sequencing machines used, and versions of sequencing software. In addition, the processing for building the model was not clearly explained, including the total amount of data used to build the model, the split of data for training and testing, and the amount and proportion of data for different groups. This information should be added to the manuscript.

Some specific comments are as follows.

Specific comments

1. Abstract

Line 32-35: Authors described “…the usage of SAMBA is not limited to aquaculture systems and can be used for modelling microbiome-host network relationships in any vertebrate organism…” 

The microbiome of each species can vary widely, as can its diet and environment. How can SAMBA be applied to other species?

2. Introduction

Line 82: “diet as the main changing experimental variable”

Why the select the “diet” as the main changing experimental variable?

3. Materials and methods

SAMBA Modules

In this section, the authors have provided very detailed information about SAMBA modules, which is commendable. However, authors should also explain how the SAMBA modules were constructed. In particular, it would be important to know the amount of data used for training and testing to build the BN (Bayesian Network) model, as well as the amount and proportion of data allocated to the training and testing phases.

Line 113: “…The three score functions are: Akaike Information Criterion (AIC), Bayesian Information Criterion (BIC), and Multinomial log-likelihood (loglik).

Apart from these few score functions, are there any other reference factors to evaluate the model? For example, a confusion matrix.

Artificial testing dataset (sequencing and dataset definition)

Line 178:”… for 16S rRNA genes using the Oxford Nanopore MinION platform…”

Authors should provide additional information on the version or chip type used for nanopore sequencing.

Is the Nanopore sequencing Full-length or V3-V4 16s rRNA? If it is not Full-length, please provide an explanation for the choice.

For line 17, it is essential to explain how the two PCR conditions were determined.

Empirical testing S. aurata dataset

Line 192: ”…the V3-V4 hypervariable regions of 16S rRNA”

What sequencing platform was used to generate this data, and would any discrepancies with the artificial test dataset potentially affect the analysis results?

Line 198: ” The resulting microbial populations were investigated in relation to different feeding scenarios”

Why select the “different feeding scenarios” as the main changing experimental variable? How about other factors (such as environment)?

4. Results and Discussion

Line 229: “…effects of genetics, environment, or traceability factors.” and

Line 241: “…. or any other environmental condition…” 

The authors mentioned that environmental conditions may provide statistically significant information, suggesting that changes in feed formulation or other environmental factors may modulate the microbiome profile. However, the study did not explain the environmental differences between the different groups and only focused on comparing different diets.

In summary: Please explain the uniqueness and importance of SAMBA and why one should choose to use SAMBA instead of developing a model independently using BN or other methods.

Round 2

Reviewer 2 Report

Additional Comments: 

Section 2.2: Artificial Testing Dataset (Sequencing and Dataset Definition)

There are some issues within this section that need to be addressed.

Firstly, the "mock community" comprises eight bacteria, not seven, as found in the ZymoBIOMICSTM Microbial Community Standard II product.

Secondly, the authors provide only a brief description of the Nanopore sequencing process for the full-length 16S rRNA (V1 to V9). However, there is a lack of information on how the raw reads were processed. This is crucial as details on basecalling and obtaining raw abundance counts for taxonomy (utilizing different databases or methods leading to varying outcomes) are absent. More comprehensive information should be provided.

Furthermore, it would be helpful if the text explains why the Artificial Testing Dataset used Nanopore sequencing, while the Empirical Testing S. aurata dataset employed the Illumina MiSeq system. Offering an explanation could shed light on the suitability of each platform for such applications.
